# Locality-aware Concept Bottleneck Model

**Sujin Jeon**[*1]    **Inwoo Hwang**[*1]    **Sanghack Lee**[†12]    **Byoung-Tak Zhang**[†1]

[1]AI Institute, Seoul National University
[2]Graduate School of Data Science, Seoul National University

## Abstract

Concept bottleneck models (CBMs) are inherently interpretable models that make predictions based on human-understandable visual cues, referred to as concepts. As obtaining dense concept annotations with human labeling is demanding and costly, recent approaches utilize foundation models to determine the concepts existing in the image. However, such *label-free* CBMs often fail to attend to concepts that are important predictive but only exist in a small region of the image (e.g., a beak of a bird), making their decision-making less aligned with human reasoning. In this paper, we propose a novel framework, coined Locality-aware CBM (LCBM), which divides an image into smaller patches. Specifically, we use their similarity with concepts to ensure that the concept prediction of the model adheres to the relevant region and effectively captures important local concepts existing in the small region of the image. Experimental results demonstrate that LCBM accurately identifies important concepts from images and exhibits improved localization capability while maintaining high classification performance.

## 1   Introduction

The interpretability of deep neural networks has become an increasingly important issue as the rapid advancements of AI make them closely integrated into our daily lives. This is especially critical in fields where the reliability of models is paramount, e.g., healthcare. Since explaining the decision-making process of a black-box model is often challenging, recent works focus on building inherently interpretable models, i.e., whose decision-making is naturally easily understandable to humans.

Concept-bottleneck model (CBM) [4, 6, 7, 17, 20] is a representative inherently interpretable architecture where a *concept* refers to human-recognizable visual cues. CBMs first predict the concepts existing in the image (e.g., "red color" and "round shape" in the image of an apple) and make a final prediction on the class label through the linear combination of these concepts. However, their reliance on dense annotations of which concepts are present in the image limits their scalability and practicality. This has led to the emergence of label-free CBMs [8, 12, 16, 21, 22], which utilize foundational models such as large language models (LLMs) and vision-language models (VLMs) to determine and estimate the presence of concepts.

However, the interpretability of label-free CBMs is often compromised due to their neglect of *locality* in two key ways. First, they often fail to focus on important local features since they analyze the entire image with VLMs to infer concept presences. This prevents the model from explaining decisions in terms of fine-grained, localized concepts, thereby limiting their interpretability. Second, it is suggested that label-free CBMs fail to properly attend to the corresponding regions in the image when predicting concepts [10, 11, 14]. This discrepancy raises concerns on their interpretability as their concept prediction does not align with the spatial locality of each concepts.

---

[*]Equal contribution.
[†]Corresponding authors.

38th Conference on Neural Information Processing Systems (NeurIPS 2024).

To this end, we propose a novel framework, coined Locality-aware CBM (LCBM). To ensure locality, we begin by dividing the image into smaller patches. To make the prediction on each concept properly attends to the corresponding region, we use the CLIP similarity between concepts and image patches to guide the training of our model, as a higher CLIP score for a concept and a patch implies a higher probability that the concept exists in the corresponding local region in the image. We experimentally demonstrated that LCBM can effectively predict concepts from small regions while also focusing on the corresponding region during the prediction process.

## 2  Related Works

**Concept Bottleneck Models (CBMs)** The concept bottleneck model (CBM) [7] is an inherently interpretable framework that explains its predictions through the *concepts*, which represent canonical visual cues composing the objects and scene. It first predicts which concepts exist in the image and then makes the final prediction based solely on these predicted concepts, enabling model decisions to be explainable in terms of concept presence. Building on this foundational idea, several works have aimed to improve its reliability [6] or accuracy-interpretability trade-off [4, 17, 20].

**Label-free CBMs** A significant drawback of conventional CBMs is that they rely on dense annotations indicating which concepts are present in the image. This demands extensive human labor, which makes them impractical to be applied to large-scale datasets like ImageNet [3]. To address this, recent label-free CBMs [8, 12, 16, 21–23] utilize foundation models to determine the concept presence without human annotation. For example, PCBM [23] leverages CLIP [13] to estimate the presence of concepts. Label-free CBM [12], LaBo [22], and LM4CV [21] additionally utilize LLMs to automate the curation of a concept set. Several studies have tackled issues related to concept faithfulness [8] and completeness [16] in such models. Yet, the problem of locality still remains unresolved.

## 3  Method

In this section, we present our novel framework whose prediction is inherently interpretable through a composition of distinct local parts and their attributes, collectively referred to as concepts. We first describe the curation of a concept set using LLM (Sec. 3.1). We then describe each component of our method, including two novel losses alongside the classification loss. These losses encourage that the appropriate region to be attended to when predicting a concept, while also promoting that the concept prediction is based on its presence in a specific local region. (Sec. 3.2). The overall architecture of our method is illustrated in Fig. 1.

### 3.1  Concept Set Generation

Here, we describe the automated process of concept generation. We employ the GPT-4-omni [2] to curate a set of concepts similar to previous label-free CBMs [12, 22]. To provide a compositional explanation, we prompt GPT to generate potential concepts for the object category (e.g., birds in the CUB dataset [18]) rather than class-specific concepts as in prior approaches. Next, we prompt GPT again to align these concepts with each class, yielding a set $\mathcal{C} = \{c_1, \cdots, c_K\}$ of up to 20 relevant concepts per class.

### 3.2  Architecture

We now describe the overall architecture of our model. For each sample $(x, y)$ where $x$ is the input image and $y$ is the corresponding label, we first extract features with an encoder $f$. This yields a feature map $F \in \mathbb{R}^{H \times W \times D}$, where $H$ and $W$ are the spatial dimensions, and $D$ is the feature dimension. We apply average pooling over the spatial dimensions to obtain $F_p$, which is then mapped to concept logits $l_c \in \mathbb{R}^K$ through the linear layer, representing the predicted presence of each concept in the image. Through the final classification layer, the model outputs the class-label logits $l_y = l_c \varphi(W_y)^T$ where $W_y$ is the weights and $\varphi$ is the activation function which ensures that the weights remain positive, as negative weights are difficult to interpret. We use the classification loss:

$$\mathcal{L}_{class} = \mathcal{L}_{CE}(\hat{y}, y), \tag{1}$$

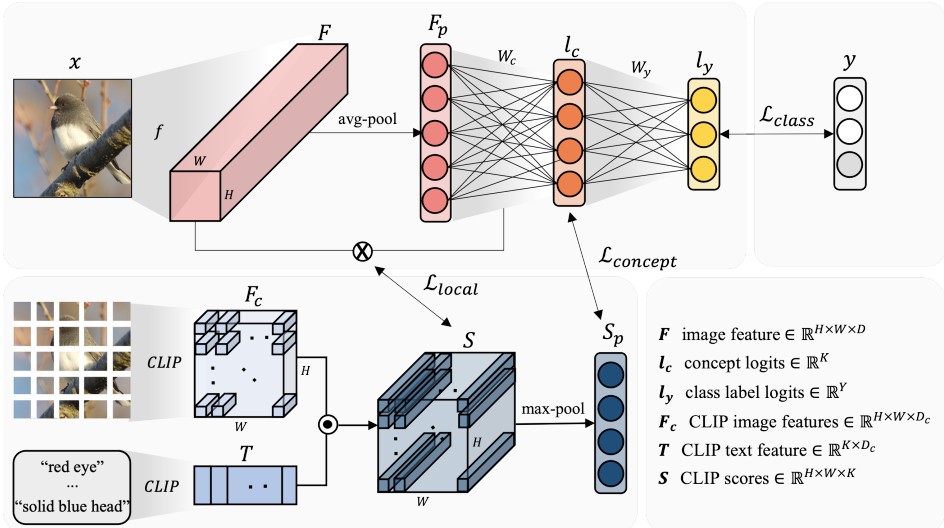

Figure 1: Overview of our method. Given an input image $x$, we first extract features $F$ and obtain concept logits $l_c$ and class label logits $l_y$ by applying linear layers to the average-pooled features $F_p$. We employ standard cross-entropy for the classification loss $\mathcal{L}_{class}$. In parallel, we crop the image into $H \times W$ patches and extract their CLIP features ($F_c$). We extract CLIP text features of the concept set, denoted as $T$. The similarity score matrix $S$ is computed as the dot product between $F_c$ and $T$, where each element $S_{(h,w,k)}$ represents the alignment between a concept $c_k$ and a patch $(h, w)$. $\mathcal{L}_{local}$ facilitates the concept prediction of the model adheres to the spatial locality of each concept. $\mathcal{L}_{concept}$ guides the model to better capture local concepts existing in the small region of the image.

where $\mathcal{L}_{CE}$ is the standard cross-entropy loss and $\hat{y}$ is the prediction of the model, i.e., softmax over the class-label logits $l_y$.

Building on top of this concept bottleneck architecture, we aim to improve its localization capabilities, i.e., to better capture the important local concepts and better align its concept prediction to the corresponding local region of the image. First, we crop the image into $H \times W$ patches and extract the CLIP features for each patch, resulting in a feature map $F_c \in \mathbb{R}^{H \times W \times D_c}$, where $D_c$ is the feature dimension. Similarly, we compute CLIP features for the concept set $\mathcal{C}$, yielding $T \in \mathbb{R}^{K \times D_c}$. We then obtain the similarity score matrix $S = \bar{F}_c \bar{T}^T \in \mathbb{R}^{H \times W \times K}$, where $\bar{\cdot}$ indicates normalized tensors. To ensure the proper localization of concepts within the image, we compute the influence value for each concept $c_k$ as:

$$V_{k,h,w} = \sum_{d=1}^{D} F_{h,w,d} \frac{1}{HW} \sum_{h'=1}^{H} \sum_{w'=1}^{W} \left[ \frac{\partial l_c}{\partial F} \right]_{k,h',w',d}. \tag{2}$$

We aim to align $V_k$, i.e., the distribution of the influence value for the concept $c_k$ over the spatial dimensions, and $S_k$, i.e., the CLIP scores between $c_k$ and all image patches, as follows:

$$\mathcal{L}_{local} = \sum_{k=1}^{K} D_{KL} \left( \sigma(V_k) \| \sigma(S_k/\tau_l) \right), \tag{3}$$

where $D_{KL}$ denotes the KL-divergence, and $\sigma$ and $\tau_l$ represent the softmax function and its temperature, respectively.

Finally, we apply max-pool over the spatial dimensions of the similarity score matrix $S$ to produce $S_p \in \mathbb{R}^K$, representing how strongly each concept $c_k$ is activated in the image. Intuitively, this allows us to effectively capture the concepts existing in the small local region of the image. We align the calculated concept logits with the actual concepts present in the image by minimizing the KL divergence between the distribution of $l_c$ and $S_p/\tau_c$, as follows:

$$\mathcal{L}_{concept} = D_{KL} \left( \sigma(l_c) \| \sigma(S_p/\tau_c) \right). \tag{4}$$

The final loss is $\mathcal{L}_{total} = \alpha\mathcal{L}_{class} + \beta\mathcal{L}_{local} + \gamma\mathcal{L}_{concept}$, where $\alpha$, $\beta$, $\gamma$ are balancing coefficients.

Table 1: Main results.

| Model | Accuracy | Precision | Recall | F1 | Localization |
|---|---|---|---|---|---|
| LfCBM [12] | $0.727_{\pm 0.001}$ | $0.146_{\pm 0.001}$ | $0.251_{\pm 0.002}$ | $0.173_{\pm 0.002}$ | $0.194_{\pm 0.006}$ |
| LaBo [22] | $0.734_{\pm 0.000}$ | $0.580_{\pm 0.001}$ | $\mathbf{0.730}_{\pm 0.000}$ | $0.626_{\pm 0.001}$ | - |
| LCBM (Ours) | $\mathbf{0.741}_{\pm 0.004}$ | $\mathbf{0.702}_{\pm 0.003}$ | $0.678_{\pm 0.013}$ | $\mathbf{0.672}_{\pm 0.006}$ | $\mathbf{0.515}_{\pm 0.007}$ |

## 4  Experiments

### 4.1  Setup

We used the CUB dataset [18] which contains 11,788 images across 200 bird species, with 5,994 images for training. Given many species pairs with small inter-class differences, accurately identifying fine-grained concepts in the images is crucial for correct classification. For the baselines, we compare our method with two representative label-free approaches, LfCBM [12] and LaBo [22]. For the evaluation, we measure the classification accuracy of the models on the validation set. We selected the concepts whose scores (i.e., $l_c \times \varphi(W_y)$) are higher than the defined threshold, and considered them as the concepts predicted to be present in the image by the models.

### 4.2  Evaluation

**Concept evaluation**   To evaluate whether the predicted concepts truly exist in the image, we calculated the precision, recall, and F1-score of the predicted concepts. Ground-truth concepts for these metrics were selected from the concepts associated with the label, defined by Sec. 3.1. Whether a concept is considered ground-truth was determined by Qwen2-VL [19], a large VLM capable of visual question answering. Specifically, we queried the VLM *"Does {concept} appear in this bird image?"* and regarded the concept as ground-truth when the VLM output positive answer. We then measured the text similarity between the predicted concepts and ground-truth concepts using OpenAI's text-embedding-3-large model [1]. Predicted concepts with a text similarity score higher than 0.8 were considered to be present in the image.

**Localization**   To evaluate the model's localization ability, we generated a GradCAM [15] based score map for each predicted concept. By applying a threshold to the score map, we extracted the most highly activated area corresponding to each concept. We then used the CUB dataset's ground truth annotations which specify the location point for each part (e.g., head). For each annotated part, if there exists corresponding predicted concept (e.g., 'striped black head' for head part), we calculated whether the activated area of matched concept contains the ground truth location point.

### 4.3  Results

**Classification accuracy**   The first column in Table 1 shows the classification accuracy. We used the same concept set generated by the LLM at 3.1 for all baselines. LCBM achieved the highest accuracy among the baselines, demonstrating that it maintains generalization ability while enhancing both concept prediction capability and localization.

**Concept evaluation**   As shown in Table 1 (precision, recall, F1), LfCBM performs poorly in these metrics as it considers only the entire image, making it difficult to discover fine-grained concepts. LaBo, which also uses only the entire image for reasoning, shows higher concept evaluation scores due to its weight prior. The weights between a class and its corresponding concepts (class-concepts) are larger than other weights, so the predicted concepts tend to heavily reflect the class-concepts, even if some are not present in the image. This is evident from the gap between LaBo's precision and recall: while recall is high because a large portion of class-concepts are included in the predictions, precision is much lower since many of these predicted concepts are not actually present in the image. In contrast, LCBM achieves high precision and comparable recall, indicating that it more accurately identifies the concepts that are genuinely present in the image.

**Locality measure**   The fifth column of Table 1 presents the average ratio of part whose activated area of corresponding concept contains its ground truth location point. This shows that LCBM has

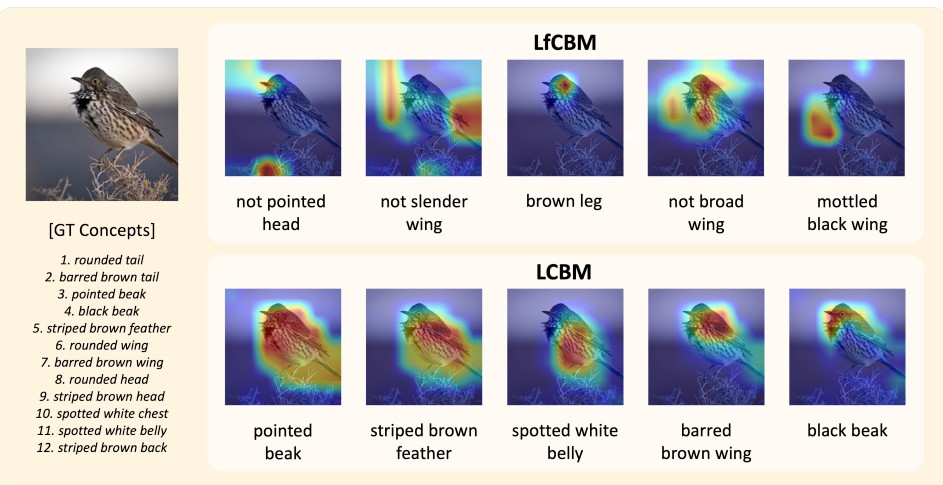

Figure 2: Qualitative result. The picture on the left shows the model input, while the concepts under denote ground truth concepts determined by Qwen2-VL. The first and second rows on the right display the localization results for the top-5 concepts predicted by LfCBM and our model, respectively. Each result includes the predicted concept along with the corresponding GradCAM based activation map, which is depicted on the image. Color closer to red means higher activation.

a significantly improved ability to correctly localize the corresponding area compared to LfCBM. LaBo is excluded from this evaluation since it does not train a concept prediction layer.

### 4.4 Qualitative analysis

Fig. 2 presents the qualitative localization results, comparing our model (LCBM) and LfCBM. The results demonstrate that our model accurately predicts concepts that are aligned with the image's class and the image content itself. More importantly, our model effectively localizes four out of the five predicted concepts, with high activation in the relevant parts of the image. For instance, when the concept contains 'belly', our model strongly activates the region where the belly is located. In contrast, LfCBM struggles with localization, showing more random behavior and failing to focus on the correct areas.

## 5 Conclusion

In this paper, we tackled the issue of neglecting locality in existing label-free CBMs by introducing a novel framework, LCBM. Our approach utilizes cropped image patches and incorporates two additional losses to ensure reliable prediction and localization of concepts. Experimental results demonstrate that LCBM significantly improves concept prediction accuracy and localization within the image.

## Acknowledgments and Disclosure of Funding

This work was partly supported by the IITP (RS-2021-II212068-AIHub/10%, RS-2021-II211343-GSAI/15%, RS-2022-II220951-LBA/15%, RS-2022-II220953-PICA/20%), NRF (RS-2024-00353991-SPARC/20%, RS-2023-00274280-HEI/10%), and KEIT (RS-2024-00423940/10%) grant funded by the Korean government.

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

# A    Experimental Details

## A.1    Baselines

Here, we provide the details of the baselines, LfCBM and LaBo.

- LfCBM [12]: LfCBM sequentially trains the concept prediction layer and classification layer, using CLIP scores with cos-cubed similarity to guide concept prediction. However, since LfCBM considers only the entire image when obtaining CLIP scores, it is affected by the previously mentioned issues of label-free CBMs.
- LaBo [22]: LaBo treats the concept prediction layer as fixed, directly using the calculated CLIP scores as input to the classification layer. Since it initializes the weights between a class and its corresponding concepts to 1, it exhibits false-positive behavior—predicting concepts that are not actually present in the image but are commonly associated with images of the same class.

# B    Implementation Details

For all models, we trained the model with a concept set defined in 3.1, which consists of a total of 211 concepts. All models utilize ViT-B/16 CLIP to extract CLIP features. Additionally, model selection for all baselines was based on the highest validation accuracy observed at each check interval during training.

## B.1    Baselines

In LfCBM [12], we omitted the concept filtering procedure to ensure that all baselines are evaluated on the same concept set. Since adding weight prior to LfCBM appeared to have no significant effect on performance due to the sparsity regularization, we followed the model structure described in the original paper without including any weight prior. We use the publicly available code provided by the authors.[3]

For LaBo [22], we omitted the concept selection module that prunes concepts to ensure that every model use the same concept set. We use the publicly available code provided by the authors.[4]

## B.2    Ours

We utilized ResNet-50 [5], pretrained on ImageNet [3], as our feature extractor and fine-tuned it during training. For the losses $\mathcal{L}_{concept}$ and $\mathcal{L}_{local}$, we cropped the image into patches of size 64 with some overlap between patches. The model was trained in an end-to-end manner with hyperparameters $\tau_l = 0.1$, $\tau_c = 0.1$. Additionally, we thresholded the CLIP score $S$ with the value set to 0.29. ReLU was used as the activation function $\varphi$, and weight prior was applied to $W_y$ similar to LaBo. We optimized our model using AdamW [9] with a learning rate of 0.0001 and $\alpha = 2$, $\beta = 1$, $\gamma = 1$. The training was conducted on four NVIDIA A100 GPUs over 20 epochs.

---

[3]https://github.com/Trustworthy-ML-Lab/Label-free-CBM
[4]https://github.com/YueYANG1996/LaBo

