# OpenReview forum: "Locality-aware Concept Bottleneck Model"
_NeurIPS.cc/2024/Workshop/UniReps — UniReps_

### Official Review · Reviewer_yY4z · 2024-10-04
**The paper proposes a locality preserving CBM model which can extract granular information from the image and making a more aligned human interpretable concept reasoner.**

**Rating:** 7
**Confidence:** 5

**Review:**

The paper has interesting ideas but it talks about the locality of the images (i.e say beak of the bird) but doesn't provide experiments to validate this. A good example would be that if we can align what the CUB attribute list is and do a ROC classification with the extracted attributes. Though the authors did a localisation task was it only covered for 'head' or other attributes too.

---

### Official Review · Reviewer_Xs2k · 2024-10-06
**Review on LCBM**

**Rating:** 6
**Confidence:** 3

**Review:**

This paper introduces the Locality-aware Concept Bottleneck Model (LCBM), aiming to address the limitations of label-free Concept Bottleneck Models (CBMs) in capturing local concepts that exist in small regions of images.

Strengths :
1. Addressing a Relevant Problem:

The paper effectively highlights the issue of locality in CBMs, where existing models often fail to capture important local details within images. By adopting a patch-based method and leveraging CLIP for concept localization, it presents a novel approach to improving the alignment between concepts and specific image regions.

2. Enhanced Localization and Interpretability:

The experimental results demonstrate improved localization, as indicated by the Intersection over Union (IoU) metric. LCBM outperforms baseline methods like LfCBM in identifying localized features such as bird beaks and heads, with a notable increase in IoU scores. By focusing on more localized concepts, the model is able to enhance transparency in decision-making.


Weakness :

1. No Justification for Patch Size and Partitioning :

The paper does not provide a clear explanation or reasoning for the chosen patch size or the method used to partition the image. Dividing images into patches and treating each one independently assumes that each concept can be isolated in a small region, which may not always hold true. There is no discussion of why the selected patch size is optimal or how sensitive the model is to this parameter. This could result in arbitrary or suboptimal choices in practice, where an incorrect patch size might misrepresent local features or overlook important visual information spread across patches.


2. Overlooking the Effect of Concept Overlap:

The method assumes that each patch contains distinct and isolated concepts, but in many real-world scenarios, visual concepts often overlap or are interdependent. By rigidly dividing images into patches, the model may fail to capture these interdependencies, leading to flawed predictions. This limitation overlooks the complexity of visual data, where multiple concepts may need to be considered together to achieve accurate predictions.

Conclusion:

Overall, this paper tackles an important problem in CBMs by focusing on locality, offering a solution through LCBM. While there are some limitations, particularly around the choice of patch size and handling concept overlap, the overall approach is promising and addresses key issues in the field. I believe the strengths outweigh the weaknesses, and this paper should be accepted.

---

### Official Review · Reviewer_N3WY · 2024-10-06
**The proposed method is straightforward, but there are some questionable parts.**

**Rating:** 5
**Confidence:** 3

**Review:**

## Summary:
To tackle the issue of label-free Concept Bottleneck Models (CBMs) neglecting essential concepts that appear in only small regions of an image, the author introduced a novel framework called Locality-Aware CBM (LCBM). This approach divides the image into smaller patches and uses CLIP to extract concept labels, measuring the similarity between these labels and image patches to guide the model's training. Experiments on the CUB dataset demonstrated reasonable classification and concept localization performance.

## Strength:
- S1: The proposed method is straightforward to understand.

## Weakness:
- W1: While the author's use of the CLIP score for training guidance appears reasonable, whether the proposed method offers an advantage over directly utilizing the CLIP or foundation model to represent the concept logit \(l_c\) is questionable. Since the proposed approach introduces an additional backbone to be trained with CLIP, it would be valuable to include additional experimental results comparing the proposed method with a pure CLIP-based configuration.

- W2: In explaining the equation for computing the class-label logits in lines 84 and 85, the author justifies using a specific activation function to ensure that the weights remain positive. However, this rationale appears inadequate, as a negative relationship between the concept and the logit can also be valid and meaningful.

- W3: The authors presented experimental results on concept evaluation and localization; however, more comprehensive experimentation is required. The CUB dataset contains approximately 300 binary concepts, which could be utilized more extensively in the concept evaluation experiments. Furthermore, the authors focused solely on the head region in the concept localization experiments, which needs to be revised and warrants broader evaluation.

---

### Official Review · Reviewer_NC5K · 2024-10-07
**Technically solid paper with clever tweak for label-free CBM**

**Rating:** 8
**Confidence:** 4

**Review:**

### Summary

This paper introduces locality-aware concept bottleneck models to improve label-free CBM by dividing the image into smaller patches and using concept similarity.

### Strengths

- The main idea of patching for locality is simple, clever, and novel.
- The results are strong, conclusive, and underline the effectiveness of the approach

### Areas for development

- This work seems closely related to concept transformers [1], which also split the images into patches. Although they require concept annotations to work but in practice you could train them to work with label-free approaches. I think it would be helpful to outline how this approach varies from this.
- It would be interesting to see an ablation on how overlapping vs. non-overlapping patches and things like patch size change the effectiveness of this method.

[1] Rigotti, Mattia, et al. “Attention-based interpretability with concept transformers.” *International conference on learning representations*. 2021.

---

### Official Review · Reviewer_A3rg · 2024-10-07
**The work presents a new and improved procedure to build Concept Bottleneck Models (CBMs) that are also sensitive to the actual location of the concepts in the original image. The work achieves improved performance in-terms of accuracy and localisation scores.**

**Rating:** 6
**Confidence:** 4

**Review:**

Summary -
The work presents a new and improved procedure to build Concept Bottleneck Models (CBMs) that are also sensitive to the actual location of the concepts in the original image. The work achieves improved performance in-terms of accuracy and localisation scores. Thus, improving the utility of CBMs by making them more location aware.

Strengths -
1. The paper tackles an important problem, of localising concepts to specific image regions for Concept Bottleneck Models.
2. The work proposes an supposedly easy formulation to incorporate alignment and make the concepts location-aware.
3. The work achieves improved performance in terms of accuracy, precision / recall / f1 scores, and localisation scores of the concepts.
4. The single qualitative example shown also demonstrates some promise of the proposed method.

Weaknesses -
1. The Formulation of the loss L_local and influence value V_k,h,w is not very clear, and the motivation for using this is not properly explained. A better explanation and motivation of this would be really helpful.
2. The final loss is a weighted sum of L_class , L_local, and L_concept; with weights alpha, beta and gamma. A proper ablation (backed by empirical results on a held-out dataset) of how these are chosen should be demonstrated.
3. Although the method demonstrates improved localisation scores as compared to prior works, however the mIOU scores of 0.202 are still very low, and it is unclear if these are even meaningful (which is also somewhat indicated by the qualitative results).
4. Finally the work only demonstrates the quantitative evaluations on as single (fine-grained birds dataset - CUB), and further evaluation across more datasets would strengthen the results.

Minor -
In the abstract, the following line is not very clear or grammatically correct and is recommended to be re-written.- "However, such label-free CBMs often fail to attend to concepts that are important predictive but only exist in a small region of the image (e.g., a beak of a bird), making their decision-making less aligned with human reasoning."

---

### Decision · Program_Chairs · 2024-10-10

**Decision:**

Accept

**Comment:**

In light of the positive reviewers' feedback and relevancy of the submission, we are pleased to accept this paper for presentation at UniReps 2024. We kindly ask the authors to incorporate the reviewers' suggestions and feedback in the final camera-ready version of the manuscript.